# Immunohistochemical Analysis Revealed a Correlation between Musashi-2 and Cyclin-D1 Expression in Patients with Oral Squamous Cells Carcinoma

**DOI:** 10.3390/ijms21010121

**Published:** 2019-12-23

**Authors:** Giuseppe Troiano, Vito Carlo Alberto Caponio, Gerardo Botti, Gabriella Aquino, Nunzia Simona Losito, Maria Carmela Pedicillo, Khrystyna Zhurakivska, Claudia Arena, Domenico Ciavarella, Filiberto Mastrangelo, Lucio Lo Russo, Lorenzo Lo Muzio, Giuseppe Pannone

**Affiliations:** 1Department of Clinical and Experimental Medicine, University of Foggia, Via Rovelli 50, 71122 Foggia, Italy; giuseppe.troiano@unifg.it (G.T.); vito_caponio.541096@unifg.it (V.C.A.C.); mariacarmela.pedicillo@unifg.it (M.C.P.); claudia.arena@unifg.it (C.A.); domenico.ciavarella@unifg.it (D.C.); filiberto.mastrangelo@unifg.it (F.M.); lucio.lorusso@unifg.it (L.L.R.); lorenzo.lomuzio@unifg.it (L.L.M.); giuseppe.pannone@unifg.it (G.P.); 2Pathology Unit, Istituto Nazionale per lo Studio e la Cura dei Tumori, “Fondazione G. Pascale”, IRCCS, 80131 Naples, Italy; g.botti@istitutotumori.na.it (G.B.); g.aquino@istitutotumori.na.it (G.A.); n.losito@istitutotumori.na.it (N.S.L.)

**Keywords:** MSI2, OSCC, oral cancer, musashi 2, prognosis

## Abstract

Aim: Musashi 2 (MSI2), which is an RNA-binding protein, plays a fundamental role in the oncogenesis of several cancers. The aim of this study is to investigate the expression of MSI2 in Oral Squamous Cell Carcinoma (OSCC) and evaluate its correlation to clinic-pathological variables and prognosis. Materials and Methods: A bioinformatic analysis was performed on data downloaded from The Cancer Genome Atlas (TCGA) database. The MSI2 expression data were analysed for their correlation with clinic-pathological and prognostic features. In addition, an immmunohistochemical evaluation of MSI2 expression on 108 OSCC samples included in a tissue microarray and 13 healthy mucosae samples was performed. Results: 241 patients’ data from TCGA were included in the final analysis. No DNA mutations were detected for the MSI2 gene, but a hyper methylated condition of the gene emerged. MSI2 mRNA expression correlated with Grading (*p* = 0.009) and overall survival (*p* = 0.045), but not with disease free survival (*p* = 0.549). Males presented a higher MSI2 mRNA expression than females. The immunohistochemical evaluation revealed a weak expression of MSI2 in both OSCC samples and in healthy oral mucosae. In addition, MSI2 expression directly correlated with Cyclin-D1 expression *(p* = 0.022). However, no correlation has been detected with prognostic outcomes (overall and disease free survival). Conclusions: The role of MSI2 expression in OSCC seems to be not so closely correlated with prognosis, as in other human neoplasms. The correlation with Cyclin-D1 expression suggests an indirect role that MSI2 might have in the proliferation of OSCC cells, but further studies are needed to confirm such results.

## 1. Introduction

Oral cancer (OC) belongs to the wider family of Head and Neck Cancers (HNCs). OC is a highly relevant problem for global public health, with a clinical impact in terms of incidence, prevalence, and mortality rates that do not tend to improve. It is reported to be the 11th most common malignancy worldwide [1]. Around 90% of OCs are histologically classified as squamous cell carcinoma (OSCC), involving the mucosal surface of the oral cavity and tongue [2]. Oral carcinogenesis encompasses multistep processes that drive the progression from normal mucosa to OSCC [3]. Changes in the DNA sequence, accumulation of somatic mutations and epigenetic events are the main mechanisms that are involved in tumor progression. In particular, epigenetic and post-transcriptional events, gained an important role in cancer [4]. Key-regulators of these mechanisms are the RNA-Binding Proteins (RBPs) that cause variations in protein expression, due to their involvement in splicing, mRNA-polyadenylation, editing, and r-tRNA stabilization [5]. Musashi-2 (MSI2) is one of the most studied RBPs. In particular, different studies evaluated its role in cancer. For example, MSI2 overexpression was linked to an increase of invasion and metastasis in non-small cell lung carcinoma, whereas its depletion showed a decrease of epithelia-mesenchymal transition [6]. In bladder cancer, the differentiation antagonizing non-protein noding RNA (DANCR) long non-coding RNA (lncRNA) acts by sponging miR-149 increasing the expression of MSI2, getting worse a malignant phenotype [7]. In addition, MSI2 seems to be involved in patients’ prognosis, resulting as prognostic factor in gastric [8] and cervical cancer [9]; in lung cancer, MSI2 emerged as a novel therapeutic target [10]. Several other factors that are responsible of the regulation of cell proliferation and cell cycle control have been proposed as diagnostic, prognostic, and therapeutic markers for certain malignancies. Among these, the cyclin D1 has been deeply investigated and were shown to be essential for the tumorigenesis of melanoma, breast cancer, and colon and oral squamous cell carcinoma (OSCC) [11]. Cyclin D1 belongs to the family of Cyclins and it is essential in the regulation of cell proliferation, DNA repair, and cell migration control [12]. The aim of this study was to investigate the expression of MSI2 in OSCC samples, through a histologic and bioinformatics analysis in order to evaluate its correlation to clinic-pathological variables and prognosis. Furthermore, a staining for Cyclin-D1 has been performed on OSCC tissue microarray (TMA) and the correlation of Cyclin D1 expression with MSI2 was investigated.

## 2. Results

### 2.1. Analysis of MSI2 Mutations, Gene Methylation and mRNA Expression in TCGA Database

A total 241 patients’ records were included in this analysis after extracting and matching clinic-pathological data from the TCGA database. Table 1 summarizes the main clinical-pathological characteristics of the included patients. DNA mutations and copy number alterations were not detected for the MSI2 gene in patients with OSCC included in the TCGA database (0/241, 0%). The expression of MSI2 mRNA (log2 (fpkm+1)) was relatively low ranging from 0.2785 to 2.7117 with a mean of 1.270734 (S.E. 0.030) and a median of 1,40673. According to the median value, patients were divided in low (≤1.40673) and high (>1.40673) MSI2 mRNA expression. Methylation status, measured in beta unit, showed a hyper methylated condition of the gene in all of the patients analyzed, the values of gene methylation ranged from 0.6802 to 0.9910 with a mean of 0.974483 (S.E. 0.0018993). The Spearman rank correlation test did not show a significant correlation between mRNA expression and methylation status of the gene (ρ = −0.44; *p* = 0.498); however, a higher methylation status was detected for the low-expression group with results that were close to the statistical significance (Mann–Whitney *p* = 0.095). MSI2 mRNA expression correlated with Grading (ρ = 0.169; *p* = 0.009) and showed a differential expression according to the gender (Mann–Whitney *p =* 0.001) with males’ samples showing a higher expression, while MSI2 methylation profile correlated to the age of patients (ρ = 0.140; *p* = 0.03) (Table 2). Univariate and multivariate analyses were performed, aiming to investigate whether MSI2 mRNA expression in the TCGA database was able to predict prognosis. The results of the univariate analysis were promising, showing a significant association between MSI2 mRNA expression (High vs low) and overall survival (Hazard Ratio, HR = 1.488; 95% C.I. 1.013–2.185; *p* = 0.045); furthermore, the results of the multivariate analysis (HR = 1.437; 95% C.I. 0.952–1.970; *p* = 0.084) were close to the threshold of statistical significance. Conversely MSI2 mRNA expression did not correlate with disease free survival (HR = 0.827; 95% C.I. 0.443–1.542; *p* = 0.549) in OSCC patients.

### 2.2. Immunohistochemical Analysis of MSI2 Expression on TMA

The IHC analysis of MSI2 protein expression was performed on a total of 108 patient’ samples included in the TMA; such patients had been treated at the National Cancer Institute “Giovanni Pascale” between 1997 and 2012. Table 3 reports the clinical pathological information of patients included in the cohort. An analysis of protein expression in the TMA samples revealed that MSI2 is not frequently expressed in OSCC, in fact 58.3% (63/108) cases analyzed resulted in being negative for MSI2 expression. Of the remaining 41.7% (45/108) samples, only 5.6% (6/108) showed higher level of MSI2 expression. The presence of MSI2 expression directly correlated with Cyclin-D1 expression (ρ = 0.279; Chi-Squared *p*-value = 0.022) (Table 4), this last one resulted to be higher expressed in males than in females (Mann–Whitney *p*-value = 0.024). The presence of MSI2 expression in the TMA cohort did not correlate with overall survival (HR = 0.575; 95% C.I. 0.278–1.190; *p* = 0.136) (Table 5). In the 13 oral healthy mucosae analyzed the expression of MSI2 was faint and mainly confined to the basal layer with a percentage of expression lower than 5% of the whole number of epithelial cells (Figure 1).

## 3. Discussion

OSCC represents more than 90% of oral cancers and it is one of the most aggressive cancers, being characterized by a mortality rate reaching 50% of patients on average [13]. Continuous efforts are made to better understand the processes that lead to its onset and progression, as well as to the discovering of potential targets for its therapy. Similarly to other solid tumors, the onset of OSCC results from the accumulation of a certain number of genetic or epigenetic alterations into the cells, which cause cell cycle dysregulation and uncontrolled cell proliferation [14] Recently, the new “omic” sciences provided a great amount of data that characterize the tumors at the molecular level and that can lead to discovering specific biomarkers that could make the tumor treatment more efficient, precise, and predictable [15]. A great contribution has been obtained from bioinformatics that allowed for analyzing this enormous amount of data, giving them a clinical significance [16].

Musashi RNA-binding protein 2 (MSI-2) has been demonstrated to be involved in several solid and blood cancers, where its expression emerged to be higher than in normal tissues and correlate with the prognosis [17]. Its role seems to be explicated in different processes, among which: epithelial-mesenchymal transition, migration, invasion, cell proliferation, and drug resistance [17]. While for many tumors, such as those arising from breast [18], cervical [9], colon [19], lung [6], etc. the role of MSI-2 proteins has been extensively studied and some target therapies proposed, no results regarding the role of MSI-2 in OSCC are reported in the literature. In this work, we combined both a bioinformatics analysis of data that were extracted from electronic TCGA database and an immunohistochemical evaluation of our samples to better understand the role of MSI-2 in oral cancer. The first interesting result emerged from the analysis of genomic data revealed that no DNA mutations and copy number alterations are detected for the MSI2 gene in patients with OSCC. A hyper methylated condition of the gene emerged in all the patients from the investigation of epigenetic modifications. The analysis of transcriptomic data showed a relatively low expression of MSI2 mRNA in the OSCC samples. However, the mRNA expression did not result to be significantly correlated to the methylation status of the gene.

The associations between its mRNA expression and some clinicopathological characteristics of patients have been analyzed to explore the clinical value of MSI2. The analysis revealed that MSI2 mRNA expression correlates with tumor grading and males show a higher level of MSI2 mRNA expression when compared to female patients. No other clinical features seemed to significantly correlate with MSI2 mRNA expression.

As to our knowledge, no difference between sexes has been highlighted in the literature until now regarding the expression of these regulating molecules. Nevertheless, in our study Males showed significantly higher expression levels of both MSI2 and Cyclin-D1. Such results emerged from the analysis of TCGA database and they have been confirmed by our samples. This could be the decisive contribution that big data can give us in the path towards increasingly personalized medicine [16].

Another question of this work aimed to investigate whether the expression of MSI2 in OSCC patients could predict the prognosis. The results emerged from correlation analysis lay for a significant association between high MSI2 mRNA expression and poor overall survival rate; meanwhile, the disease free survival seems not to be correlated with the MSI2 mRNA expression level.

We performed the immunohistochemical evaluation of healthy mucosae and OSCC samples to determine the expression levels of MSI2, in parallel with this bioinformatic analysis. It was of great interest to discover that the protein expression of MSI2 was low or even absent both in healthy samples and OSCC TMA and it did not correlate with any prognostic behavior. These data, combined with those deriving from TCGA analysis, lead us to affirm that, unlike many other cancers [8,20,21], for OSCC the expression of MSI2 appears to be a poor prognostic biomarker.

In addition to these results, some information regarding the potential biological functions of MSI2 in oral cancer emerged from the evaluation of the TMAs. In particular, the presence of MSI2 expression directly correlated with Cyclin-D1 expression. Cyclin-D1 is a protein that plays a crucial role in cell cycle regulation, including cell proliferation and growth, as well as DNA repair and cell migration control [22]. Its key role in tumorigenesis of several tumors, among which oral cancer, has been proposed [12], and a poor prognosis correlated to its overexpression [23]. Several mechanisms of cyclin D1 overexpression in OSCC have been identified. They range from amplification to polymorphisms and mutational events involving the oncogene CCND1, but a fundamental role of some signaling pathway intermediaries has been also suggested [24]. Zhang et al. [25] demonstrated in their study how MSI2 silencing inhibited leukemic cell growth and caused a decreasing of Cyclin D1 expression. Han et al. [26] drew the same conclusions, showing how the MSI2 silencing induced cell cycle arrest in G0/G1 phase, with decreased Cyclin D1 and increased p21 expression. In the same way, a study investigating the role of MSI2 in Hematopoietic stem cell activity discovered a close correlation between the expression profile of MSI2 and that of Cyclin D1 [27]. Given the results of our study, in a similar manner, MSI2 could affect the Cyclin D1 expression in the cells of OSCC, but further studies are needed to affirm this.

## 4. Methods and Materials

### 4.1. Analysis of MSI2 Expression and Methylation in The Cancer Genome Atlas (TCGA)

The gene expression RNAseq data HTSq-Fragments Per Kilobase Million (FPKM) were downloaded from UCSC Xena Browser (https://xena.ucsc.edu/) [28]. Data were downloaded for MSI2, MKI67, and CCND1 mRNA expression. This platform was also used to access and download the methylation profile quantification for MSI2 (https://gdc.xenahubs.net/download/TCGA-HNSC/Xena_Matrices/TCGA-HNSC.methylation450.tsv.gz; Full metadata—Illumina Human Methylation 450 expressed as beta unit). Data were organized in Microsoft Excel sheet and then pasted in cBioPortal for Cancer genomics in order to display visually the patients’ profile (http://www.cbioportal.org) [29]. Clinic-pathological and follow-up information were downloaded from Genomic Data Commons (GDC) Data Portal (https://portal.gdc.cancer.gov/) [30].

### 4.2. Immunohistochemistry of MSI2 Expression in OSCC Tissue Microarray

The reporting recommendations for tumor marker prognostic studies (REMARK guidelines) [31] were taken as a reference for carrying out this study. All of the patients filled written informed consent for the use of their samples, according to the institutional regulations and the ethics committee of the National Cancer Institute “Giovanni Pascale”, as “Bio-Banca Istituzionale BBI” Deliberation NO. 15 del 20 Jan. 2016, approved and registered the study. We decided to exclude patients with HPV-positive tumors and those arising from the base of the tongue, tonsils, oropharynx, and lips. Patients with a follow-up lower than eight months were also excluded. A total of 122 patients were included in this study. Of them, 103 reported follow-up information (from 8 to 150 months—mean of 47.34 S.D. 34.609), meanwhile the microarray tissue was not evaluable for 14 patients. None of the patients had undergone treatments prior to tissue collection. The patients were diagnosed of OSCC and 7^th^ American Joint Committee on Cancer (AJCC) staging system was applied. An evaluation of 13 mucosae samples from healthy subjects has been also performed. The paraffin blocks were cored in a 0.6 mm support (area of 0.28 mm^2^) and then transferred to the recipient master block while using Galileo TMA CK 3500^®^ Tissue Microarrayer. As the control, we used an H&E staining of a 4-μm TMA section. Immuno-histochemical staining was performed by using a mouse monoclonal antibody (Ab), which was supplied by abcam (Mouse monoclonal Anti-MSI2 antibody [OTI2F10]—ab156770), in addition staining for Cyclin-D1 (Ventana-Roche, SP4-R) and Ki-67 (Ventana-Roche) was also performed. We used an automated staining device (Ventana-Roche), with a streptavidin-biotin horseradish peroxidase technique (LSABHRP), in order to uncover the primary Abs. An optical microscope (OLYMPUS BX53, at ×200) detected immune-stained spots in four high power fields (HPFs) and they were analyzed by ISE TMA Software (Integrated System Engineering, Milan, Italy). Two of the authors (GP and GT) performed the observational quantification analysis in a joint session. Detre S. et al. method was applied to assess the scoring of immunostaining [32]. The intensity (I) of expression was scored from 0 to 3 (0 = no staining; 1 = yellow; 2 = light brown; and, 3 = black brown/black). The relative number of the positive stained cells (%) was scored from 0 to 4 (0 = 0%; 1 < 10%; 2 = 10-50%; 3 = 51-80%; 4 > 80%). All of the samples resulted in only being stained in the cytoplasm. We decided to categorize patients in negative/positive tumors because of the relative low expression of MSI2 in OSCC.

### 4.3. Statistical Analysis

SPSS statistical software 21.0 was used to perform all of the statistical analyses. Spearman rank correlation analysis was performed to investigate the correlation of MSI2 expression to clinic-pathological variables. We decided to use non-parametric tests because of the non-normal distribution of the variable checked by means of the Shapiro–Wilk normality test. For such reason, the Mann–Whitney test was applied to explore the difference in expression between groups. Univariate survival analysis was performed with the Kaplan–Meier method to estimate both overall survival rates and disease-free survival, while comparing results between groups with the log-Rank test. A multivariate Cox Proportional Hazard Model was built in order to assess the prognostic significance of MSI2 expression after adjusting for covariates, including the clinic-pathological variables: age, grading, staging (8th AJCC edition), and gender, as covariates.

## 5. Conclusions

The role of MSI2 expression in OSCC seems to be not so closely correlated with prognosis, as in other solid and blood tumors. The MSI2 mRNA expression in oral cancer is higher in males and it is correlated with tumor grade. The protein expression of MSI2 in OSCC samples is relatively low, but significantly correlated with Cyclin D1 expression. Further studies are necessary to investigate its role in OSCC genesis, progression, and prognosis.

## Figures and Tables

**Figure 1 ijms-21-00121-f001:**
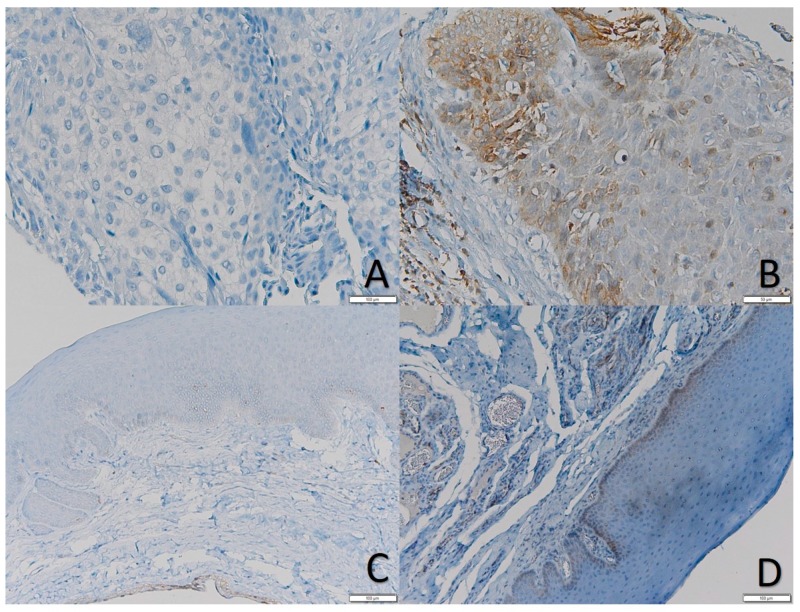
(**A**) Negative OSCC for MSI2; (**B**) Basal MSI2 positivity intensity +2; (**C**) 40% Normal mucosa negative for MSI2 with mild positivity in a ductal epithelium; and, (**D**) normal mucosa mild positive for MSI2 in the basal layer.

**Table 1 ijms-21-00121-t001:** Clinical-pathological characteristics of patients included in The Cancer Genome Atlas (TCGA) analysis.

Clinic-Pathological Information	Groups	Number of Patients
**Age**	≤ 65 years old	144/241
	> 65 years old	97/241
**Gender**	Male	158/241
	Female	83/241
**Grade**	1	43/241
	2 3	147/241 51/241
**Stage**	1–2	76/241
	3–4	165/241
**Subsite**	Tongue	103/241
	Gingivo-buccal	30/241
	Floor of the mouth	46/241
	Others	62/241

**Table 2 ijms-21-00121-t002:** Spearman rank correlation for the 241 Oral Squamous Cell Carcinoma (OSCC) patients included in the TCGA database. * *p <* 0.05; ** *p <* 0.001.

Variable	Age	Grade	Stage	Gender	Perineural Invasion	MSI2 Methylation	MSI2 mRNA Expression	Ki-67 mRNA Expression	Cyclin-D mRNA Expression
**Age**	ρ = 1	0.088	−0.084	**0.243**	0.060	**0.140**	−0.121	−0.069	0.033
	*p*-value = 1	0.175	0.197	0.001 **	0.414	**0.03 ***	0.06	0.287	0.607
**Grade**		ρ = 1	−0.003	−0.066	0.100	−0.094	0.169	−0.105	0.053
		*p*-value = 1	0.959	0.307	0.176	0.146	0.009 **	0.105	0.418
**Stage**			ρ = 1	−0.026	0.199	−0.031	−0.035	0.038	−0.062
			*p*-value = 1	0.686	0.006 **	0.632	0.587	0.561	0.340
**Gender**				ρ = 1	−0.090	−0.045	−0.220	0.042	0.043
				*p*-value = 1	0.907	0.485	0.001 **	0.520	0.505
**Perineural Invasion**					ρ = 1	−0.178	0.033	−0.132	−0.038
					*p*-value = 1	0.014 *	0.650	0.071	0.606
**MSI2 Methylation**						ρ = 1	−0.044	0.094	0.027
						*p*-value = 1	0.498	0.144	0.673
**MSI2 mRNA expression**							ρ = 1	−0.093	0.003
							*p*-value = 1	0.150	0.963
**Ki-67 mRNA expression**								ρ = 1	0.253
								*p*-value = 1	0.000 **
**Cyclin-D mRNA expression**									ρ = 1
									*p*-value = 1

**Table 3 ijms-21-00121-t003:** Clinical-pathological characteristics of patients included in the immunoistochemical analysis.

Clinic-Pathological Information	Groups	Number of Patients
**Age**	≤ 65 years old	45/108
	> 65 years old	63/108
**Gender**	Male	79/108
	Female	29/108
**Grade**	1 2	22/108 50/108
	3	36/108
**Stage**	1–2	35/108
	3–4	73/108
**Subsite**	Tongue	67/108
	Gingivo-buccal	23/108
	Floor of the mouth	13/108
	Others	5/108

**Table 4 ijms-21-00121-t004:** Spearman rank correlation for patients included in the TMA. Age. Cyclin-D1 and Ki-67 were included as continuous variables. while Grade. Stage. Gender. and MSI2 as categorical variables. * *p* < 0.05; ** *p* < 0.001.

Variable	Age	Grade	Stage	Gender	MSI2 Expression (Neg/Pos)	Cyclin-D1 Expression	Ki-67 Expression
**Age**	ρ = 1	0.049	−0.151	−0.062	−0.006	0.021	0.148
	*p*-value = 1	0.617	0.122	0.528	0.955	0.865	0.226
**Grade**		ρ = 1	0.098	0.034	0.223	0.060	0.398
		*p*-value = 1	0.313	0.726	−0.066	−0.044	0.001 *
**Stage**			ρ = 1	−0.008	0.497	0.721	−0.004
			*p*-value = 1	0.993	0.676	0.468	0.977
**Gender**				ρ = 1	−0.088	−0.277	−0.285
				*p*-value = 1	0.364	0.023 *	0.031 *
**MSI2 expression (Neg/Pos)**					ρ = 1	0.279	0.122
					*p*-value = 1	0.022 *	0.315
**Cyclin-D1 expression**						ρ = 1	0.485
						*p*-value = 1	0.000 **
**Ki-67 expression**							ρ = 1
							*p*-value = 1

**Table 5 ijms-21-00121-t005:** Multivariate Cox regression analysis for MSI2 adjusted for other clinic-pathological parameters. * *p* < 0.05.

	Overall Survival		
Variables	Hazard Ratio	95.0% C.I.	*p*-Value
Gender	1.904	0.964–3.759	0.064
Grade	1.162	0.706–1.911	0.555
Age	1.018	0.986–1.052	0.274
Stage	2.968	1.844–4.779	0.000 *
MSI2 (Pos/Neg)	0.621	0.299–1.288	0.201

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
