# Peer review of "Immunohistochemical Analysis Revealed a Correlation between Musashi-2 and Cyclin-D1 Expression in Patients with Oral Squamous Cells Carcinoma"

_ijms, 2019, doi:10.3390/ijms21010121_

Round 1
Reviewer 1 Report
In the manuscript "Musashi-2 correlates with Cyclin-D1 and its mRNA is higher expressed in males with oral squamous cell" the authors find no connection between the RNA binding protein Musahi-2 and oral squamous cell carcinoma.
While potentially interesting, the manuscript as presented can be described at best as preliminary data. The manuscript is made up of mostly negative data. While the authors provide two points that are potentially interesting, such as the sex selective over-expression of Musashi-2 and how Musashi-2 correlates with Cyclin-D1, there are no studies to determine the function significance of these observations. Even a simple knock down experiment to determine if loss of Musashi-2 has any affect on Cyclin-D1. Much more work is required to bring this manuscript up to publication standards.
Author Response
Thank you for your valuable contribution and comment.
Ragarding the function significance of the discovered correlation between MSI-2 and Cyclin-D1, there are some studies in literature (i.e.: ZHANG H, TAN S, WANG J et al. Musashi2 modulates K562 leukemic cell proliferation and apoptosis involving the MAPK pathway. Exp Cell Res 2014; 320: 119-27.; HAN Y, YE A, ZHANG Y et al. Musashi-2 Silencing Exerts Potent Activity against Acute Myeloid Leukemia and Enhances Chemosensitivity to Daunorubicin. PLoS One 2015; 10: e0136484; HOPE KJ, CELLOT S, TING SB et al. An RNAi screen identifies Msi2 and Prox1 as having opposite roles in the regulation of hematopoietic stem cell activity. Cell stem cell 2010; 7: 101-13) that show a direct involvement of Cyclin D1 in MSI2 pathway. The results of such studies have been added now in the Discussion section.
Reviewer 2 Report
The manuscript ijms-658787-peer-review-v1 entitled “Musashi-2 correlates with Cyclin-D1 and its mRNA is 3 higher expressed in males with oral squamous cells 4 carcinoma” in this manuscript. The topic is interesting, however, the manuscripts is required a major revision to be accepted for publication. In this manuscript, author should describe about Cyclin-D1 and its mRNA is higher expressed in males’ briefly and why Female expression low. The Introduction, result, Material and method section has lots of spacing and font changes are found, the author should explain why male only expressed MSI2. Moreover, the author should describe the age of patients clearly because results not clear. First of all, to go into the details the authors should mark the pages and line numbers. It is incomplete in some sentences and has some inconsistency parts. I would be happy to review again after the manuscript has been modified. However, just as examples, below comments, should be answered and covered properly in the revised one.
Below I listed some of my comments:
Abstract
L 29, 32 - has lots of spacing and font changes are found
Introduction part
The author should be explained Cyclin-D1 gene role, introduction part and abstract part there was no detail information. Author should be writing the introduction part more briefly because all sentences short and inconsistently
L – DANCR explain the abbreviation when appears in the first time
L 52, 53 – reference has as lots of spacing and font changes are found.
Results
Analysis of MSI2 mutations, gene methylation and mRNA expression in TCGA database. This part of the results author should be give more clearly Details of the patients sample because simply narrative results.
L 59, 61 – total number of patients 241 but detected patients database (0/240, 0%). Author should be checked this sentence’
L 66, 67, 72, 74 - has lots of spacing are found
L 82- (TMA) explain the abbreviation when appears in the first time
Figure and table parts;
L 93- Table 1, (Spearman rank correlation for the 251 OSCC patients) Author don’t have mentioned the patients details in the materials and methods but has presents in Table, sould give more clearly patients details.
Discussion
L 116, 123 - has as lots of spacing and font changes are found.
Methods and materials
L 163,166 - has as lots of spacing and font changes are found
Author Response
Dear Reviewer,
thank you for your valuable contribution and comments.
We have revised the manuscript according to your suggestions.
The Cyclin-D1 functions and relevance in cancer have been presented in Introduction section.
All Spacings and font changes have been corrected.
The difference between Genders has been discussed.
The age of patients has been specified.
A table with Clinical-pathological characteristics of TCGA included patients has been added. (Table 1)
All the abbreviations have been explained when used for the first time.
Round 2
Reviewer 1 Report
In manuscript is a revision of "Musashi-2 correlates with Cycline-D1 and its mRNA is expressed in males with oral squamaous cells carcinoma".
This manuscript is minimally changed for the previous version. The authors added discussion to respond to the previous reviewer comments about adding some type of functional connection between Cyclin-D1 and Musashi-2. Yet, this discussion focuses on the connection between these two proteins in other tumors, and thus there is no demonstration that this connection occurs in Oral squamous cell carcinoma, which demonstrates a lack of rigor to the experimental design.
Author Response
Dear Reviewer,
Thank you for your effort to improve our manuscript.
Since we are unable, at the moment, to perform the specific experiment you suggested, we modified the Title of the manuscript in "Immunohistochemical analysis revealed a correlation between Musashi-2 and Cyclin-D1 expression in patients with oral squamous cells carcinoma." as suggested by Editor.
In this way, it is specifically indicated that the results emerged from an Immunohistochemical evaluation.
Please find attached the new version of the manuscript.
Best regards,
Khrystyna Zhurakivska
